# The Severity of COVID-19 Affects the Plasma Soluble Levels of the Immune Checkpoint HLA-G Molecule

**DOI:** 10.3390/ijms23179736

**Published:** 2022-08-27

**Authors:** Jéssica F. C. Cordeiro, Talita M. Fernandes, Diana M. Toro, Pedro V. da Silva-Neto, Vinícius E. Pimentel, Malena M. Pérez, Jonatan C. S. de Carvalho, Thais F. C. Fraga-Silva, Camilla N. S. Oliveira, Jamille G. M. Argolo, Augusto M. Degiovani, Fátima M. Ostini, Enrico F. Puginna, João S. da Silva, Isabel K. F. M. Santos, Vânia L. D. Bonato, Cristina R. B. Cardoso, Marcelo Dias-Baruffi, Lúcia H. Faccioli, Eduardo A. Donadi, Carlos A. Sorgi, Ana P. M. Fernandes

**Affiliations:** 1Departamento de Enfermagem Geral e Especializada, Escola de Enfermagem de Ribeirão Preto—EERP, Universidade de São Paulo—USP, Ribeirão Preto 14040-902, Brazil; 2Programa de Pós-Graduação em Reabilitação e Desempenho Funcional, Faculdade de Medicina de Ribeirão Preto—FMRP, Universidade de São Paulo—USP, Ribeirão Preto 14040-900, Brazil; 3Fiocruz-Bi-Institucional Plataforma Medicina Translacional, Faculdade de Medicina de Ribeirão Preto—FMRP, Universidade de São Paulo—USP, Ribeirão Preto 14040-030, Brazil; 4Departamento de Análises Clínicas, Toxicológicas e Bromatológicas, Faculdade de Ciências Farmacêuticas de Ribeirão Preto—FCFRP, Universidade de São Paulo—USP, Ribeirão Preto 14040-903, Brazil; 5Programa de Pós-Graduação em Imunologia Básica e Aplicada—PPGIBA, Instituto de Ciências Biológicas, Universidade Federal do Amazonas—UFAM, Manaus 69080-900, Brazil; 6Departamento de Bioquímica e Imunologia, Faculdade de Medicina de Ribeirão Preto—FMRP, Universidade de São Paulo—USP, Ribeirão Preto 14040-900, Brazil; 7Departamento de Química, Faculdade de Filosofia, Ciências e Letras de Ribeirão Preto—FFCLRP, Universidade de São Paulo—USP, Ribeirão Preto 14040-901, Brazil; 8Hospital Santa Casa de Misericórdia de Ribeirão Preto, Ribeirão Preto 14085-000, Brazil; 9Escola de Educação Física e Esportes de Ribeirão Preto—EEFERP, Universidade de São Paulo—USP, Ribeirão Preto 14040-907, Brazil; 10Departamento de Clínica Médica, Faculdade de Medicina de Ribeirão Preto—FMRP, Universidade de São Paulo—USP, Ribeirão Preto 14040-900, Brazil

**Keywords:** SARS-CoV-2, COVID-19, HLA-G, cytokine, comorbidities

## Abstract

The non-classical histocompatibility antigen G (HLA-G) is an immune checkpoint molecule that has been implicated in viral disorders. We evaluated the plasma soluble HLA-G (sHLA-G) in 239 individuals, arranged in COVID-19 patients (*n* = 189) followed up at home or in a hospital, and in healthy controls (*n* = 50). Increased levels of sHLA-G were observed in COVID-19 patients irrespective of the facility care, gender, age, and the presence of comorbidities. Compared with controls, the sHLA-G levels increased as far as disease severity progressed; however, the levels decreased in critically ill patients, suggesting an immune exhaustion phenomenon. Notably, sHLA-G exhibited a positive correlation with other mediators currently observed in the acute phase of the disease, including IL-6, IL-8 and IL-10. Although sHLA-G levels may be associated with an acute biomarker of COVID-19, the increased levels alone were not associated with disease severity or mortality due to COVID-19. Whether the SARS-CoV-2 *per se* or the innate/adaptive immune response against the virus is responsible for the increased levels of sHLA-G are questions that need to be further addressed.

## 1. Introduction

Coronavirus disease (COVID-19), caused by the new coronavirus (SARS-CoV-2), exhibited a rapid worldwide spread. The clinical response of people infected is broad, encompassing asymptomatic infection, mild/moderate upper respiratory tract disease, and severe viral pneumonia with respiratory failure and even death [1,2,3]. Effective host innate and adaptive antiviral responses against SARS-CoV-2 include the production of various pro-inflammatory cytokines and the activation of CD4 and CD8 T cells, which are essential mechanisms for the control of viral replication, virus spread, inflammation, and removal of infected cells [4,5]. In patients with COVID-19, the level of helper T cells (CD3 and CD4), cytotoxic suppressor T cells (CD3 and CD8), and regulatory T cells are below the normal background [4,6]. Considering that the human immune system differs from person to person due to a large number of histocompatibility leukocyte antigen (HLA) alleles, the high diversity of the immune response represents an evolutionary strategy for humans [7,8].

The non-classical class I HLA-G has been considered an immune checkpoint molecule, whose function has been related to the regulation of the cells of the innate and adaptive immune system [9]. Its physiological expression has been detected in immune-privileged tissues such as the placenta, cornea, and thymus. On the other hand, its non-physiological expression has been described in inflammatory conditions, viral infections, and tumors [10]. As an immunosuppressive molecule, HLA-G inhibits the cytotoxic action of natural killer (NK) cells and cytotoxic T lymphocytes. This characteristic gives HLA-G a beneficial modulatory effect on pregnancy, autoimmune and inflammatory diseases, and negative effects on cancer and viral infections [11]. The increase in soluble HLA-G (sHLA-G) levels results in a decreased activity of the immune response through their binding to inhibitory receptors [12,13].

Several lines of evidence support the role of HLA-G in COVID-19 pathogenesis, including (i) the increased levels of sHLA-G molecules in patients exhibiting various clinical manifestations of the disease [14,15,16,17,18], (ii) the increased number of immune system cells expressing HLA-G [16,19], (iii) the association of the *HLA-G* 3′untranslated region polymorphism with susceptibility to the disease [18], (iv) the increased tissue expression of HLA-G in case reports [20,21], and (v) the induction of the expression of hub genes, including *HLA-G*, in lung tissue infected by SARS-CoV-2 [22].

Thus, the quantification of systemic levels of sHLA-G may represent a new and promising biomarker in COVID-19. In this study, to understand the association between HLA-G levels and COVID-19 clinical, therapeutical variables, comorbidities, and the relationship between sHLA-G and cytokine levels, we studied COVID-19 patients stratifying to distinct clinical disease severity, distinct disease treatment regimens, and different comorbidities.

## 2. Results

### 2.1. Sociodemographic, Clinical, and Laboratory Characteristics of Study Participants

We included 239 individuals in our cohort. Among those who were COVID-19 positive (189 patients), 60 patients were treated at home, and 129 were hospitalized. As a healthy control group, 50 subjects without comorbidities were included. The major demographic, clinical, laboratory, and treatment features of the studied groups are shown in Table 1, demonstrating that (i) patients with COVID-19 were older than healthy controls; (ii) most hospitalized patients were men; (iii) the majority of patients treated at home were women, exhibiting the mild form of the disease; (iv) smoking history, systemic arterial hypertension, cardiovascular disease, diabetes mellitus, and neurological diseases were more frequently observed in hospitalized patients; (v) the hemoglobin levels and the lymphocyte counts were decreased in patients, whereas the leukocyte and neutrophil counts and the neutrophil/lymphocyte ratio were increased in the whole group of patients; (vi) hospitalized patients were primarily treated with glucocorticoids, azithromycin, and ceftriaxone.

### 2.2. Plasma Levels of the sHLA-G Molecule

We quantified the plasma levels of sHLA-G (ng/mL) in the peripheral blood of the healthy control (*n* = 50) and home (*n* = 60) and hospital (*n* = 129) care groups with COVID-19. Compared with healthy controls, COVID-19 patients (whole group or stratified into undergoing home or hospital care) exhibited increased sHLA-G levels (*p* < 0.0001, for each comparison). The comparison of sHLA-G levels between the COVID-19 groups (home and hospital) revealed no significant difference. sHLA-G levels continued to be increased when compared with controls even after the stratification of patients according to age and gender, except for home care patients who represented a very small group, encompassing only three patients. No significant differences were observed between patients (home or patient care) after stratification by age and gender (Figure 1).

We analyzed the levels of sHLA-G concerning the comorbidities presented by the studied participants. The comparisons of sHLA-G levels between healthy controls with COVID-19 patients followed up at home or in the hospital, stratified according to the underlying comorbidity (hypertension, diabetes mellitus, cardiovascular disease, smoking habit, body mass index (BMI), and neurological disorder) showed no significant differences (Figure 2).

After observing that the concentration of plasma sHLA-G was not different between the groups exhibiting COVID-19 either at home or in hospital care, we stratified patients according to disease severity into asymptomatic/mild (*n* = 39), moderate (*n* = 56), severe (*n* = 32) or critical (*n* = 62) groups (Appendix A). Irrespective of disease severity, all COVID-19 clinical forms exhibited increased levels of sHLA-G when compared with controls (*p* < 0.001 for all comparisons). The highest concentrations of sHLA-G were detected in patients with the severe form of COVID-19. On the other hand, the plasma levels of sHLA-G in critical patients were significantly lower compared with severe patients (*p* = 0.0215) (Figure 3A).

Furthermore, we also stratified the patients by clinical outcomes (recovery with hospital discharge or death) and analyzed the concentration of sHLA-G in these two groups. Thus, we observed a high level of sHLA-G in both groups of patients outcome compared with the healthy controls (*p* < 0.0001), as well as in those who died compared with recovered and were discharged from the hospital (*p* = 0.0150) (Figure 3B). However, when we performed a longitudinal analysis, there were no differences in sHLA-G levels when comparing the patients in the admission to the hospital with in the discharged time (*p* = 0.9264) or died (*p* = 0.1950) (Figure 3C).

After verifying the profile of sHLA-G concerning the severity of the disease, we correlated the levels of sHLA-G with the pharmacological treatment prescribed for patients hospitalized with COVID-19. Glucocorticoid drugs, azithromycin, oseltamivir, and chloroquine/hydroxychloroquine did not seem to interfere with the sHLA-G plasma levels, since no statistical differences were observed between these groups. However, the patients treated with ceftriaxone exhibited high plasma levels of sHLA-G compared with those who did not use this medication (*p* = 0.0148) (Figure 4). The home care COVID-19 patients were not included in these analyses since their pharmacological treatment differed from that of the hospital care COVID-19 patients; they were restricted to antipyretics and analgesic drugs.

Then, we performed a Spearman correlation test matrix between the plasma levels of sHLA-G with the variables age, BMI, days of infection and days of hospitalization, clinical score, infused O_2_ concentration, international normalized ratio (INR), total leukocytes, neutrophils, lymphocytes, neutrophil–lymphocyte Ratio (NLR), glycemia, C-reactive protein (CRP), and the cytokines IL-12, TNF, IL-10, IL-6, IL-1β, and IL-8. To interpret the magnitude of correlations, the classification of correlation coefficients was adopted as follows: <0.4 (weak correlation), >0.4 to <0.5 (moderate), and > 0.5 (strong), as detailed in Appendix A [23].

The plasma levels of sHLA-G showed positive correlations with age (r = 0.3; *p* < 0.0001); BMI (r = 0.1; *p* = 0.0271); days of hospitalization (r = 0.2; *p* < 0.0001); clinical score (r = 0.4; *p* < 0.0001); infused O_2_ concentration (r = 0.3; *p* < 0.0001); total leukocytes count (r = 0.1; *p* = 0.0271); neutrophils count (r = 0.2; *p* = 0.0101); NLR (r = 0.2; *p* = 0.0107); IL-10 (r = 0.3; *p* < 0.0001); IL-6 (r = 0.3; *p* < 0.0001); and IL-8 levels (r = 0.4; *p* < 0.0001), and a negative correlation with lymphocytes count (r = -0.2; *p* = 0.0101) (Figure 5).

Regarding the clinical severity score (0–15 according to the references [24,25,26,27,28]) of the patients, sHLA-G levels exhibited positive correlations with age (r = 0.6; *p* < 0.0001); BMI (r = 0.3; *p* < 0.0001); days of infection (r = 0.4; *p* < 0.0001); days of hospitalization (r = 0.8; *p* < 0.0001); infused O_2_ concentration (r = 0.9; *p* < 0.0001); total leukocytes count (r = 0.3; *p* < 0.0001); neutrophils count (r = 0.7; *p* < 0.0001); NLR (r = 0.7; *p* < 0.0001); glycemia (r = 0.6; *p* < 0.0001); CRP (r = 0.3; *p* = 0.0012); IL-10 (r = 0.6; *p* ≤ 0.0001); IL-6 (r = 0.7; *p* < 0.0001); IL-1β (r = 0.1; *p* = 0.0176); and IL-8 levels (r = 0.7; *p* < 0.0001), and a negative correlation with lymphocytes count (r = -0.7; *p* < 0.0001) (Figure 5).

To evaluate the risk of the sHLA-G levels to determine the morbidity/outcome of COVID-19, we performed a multivariate binomial logistic regression analysis, detailed in Appendix A. We observed that sHLA-G levels, individually, did not appear to be associated with COVID-19 severity (OR: 1.0 (CI 0.99–1.00) *p* = 0.739) or with mortality (OR: 1.0 (CI 1.0–1.0) *p* = 0.015) due to COVID-19 in the presence of confounding variables, including age, gender, comorbidity, blood glucose and NLR (Figure 6).

## 3. Discussion

In this study, we presented an overview of the sHLA-G plasma levels, stratified according to the demographic and clinical parameters of individuals infected with SARS-CoV-2. First, we showed that the sHLA-G levels observed in COVID-19 patients were increased when compared with healthy controls. After stratifying patients according to the clinical severity of the disease, we observed that all degrees of severity exhibited increased sHLA-G levels compared with healthy controls. sHLA-G concentrations exhibited a gradual increase as far as the severity of the disease progressed. However, unexpectedly, patients with critical COVID-19 showed reduced levels of sHLA-G compared with patients with severe COVID-19, in agreement with a previous study [14]. These data suggest the exhaustion of the immune system.

HLA-G presents suppressive effects on the immune system, as evidenced by (i) the HLA-G ligation to inhibitory receptors present on target cells, (ii) the indirect HLA-G-induced immunosuppression caused by multiple intercellular transfers of HLA-G from HLA-G-bearing cells to neighboring HLA-G negative cells, (iii) the induction of immune effector cells (T cells, NK cells, and monocytes) into suppressor cells, and (iv) HLA-G transference by exosomes to long-distance cells, impairing the functions of immunocompetent cells [29]. Indeed, it has been proposed that the exhaustion of the immune system is associated with some major conditions, including the observation that critically ill COVID-19 patients have several dysregulations in their immune system responses, such as lymphopenia, cytokine storm, an increased frequency of depleted CD4+ and CD8+ lymphocytes [30,31], and the upregulation of the NKG2A/CD94 receptor favoring the HLA-G ligation to this receptor that may also cause of the immune exhaustion, as seen in severe disease, which is highly dependent on key pathogen-induced cytokines [32]. Therefore, during viral infections, several factors concur for the exhaustion of T cells through the progressive dysfunction of T cells due to the overexpression of inhibitory receptors of immune control [33].

Besides lymphopenia and immune exhaustion, other underlying factors have been associated with COVID-19 severity and morbidity, including age, gender and comorbidities [34,35]. In this study, COVID-19 patients undergoing home care were younger than those hospitalized; however, older age has been reported to be a risk factor for the severity of the disease [14,15,36,37]. To test the hypothesis that sHLA-G levels may be affected by age, patients at home or in hospital care and healthy controls were grouped into individuals older or younger than 60 years. We observed that patients older than 60 years did not show a significant increase in their sHLA-G levels compared with younger COVID-19 patients, indicating that age did not affect the patients’ sHLA-G levels. This result contrasts with a previous study that reported that sHLA-G levels were positively correlated with age [15]. Regarding gender, 67.4% of the people belonging to the group of hospitalized patients were men; this can be explained by the greater male vulnerability to COVID-19, as the circulating levels of ACE2 (receptor for SARS-CoV-2) in men are greater than its levels in women [38,39]. A study carried out with our population observed that testosterone, for “non-severe” disease in men, could predict the evolution of COVID-19 to worse prognoses, including death [40]. Immunosenescence could lead to an inefficient immune response to COVID-19 [41]. Comorbidities such as hypertension, type 2 diabetes mellitus, and cardiovascular diseases, when associated with aging, may increase susceptibility to COVID-19 [42]. Although the hospitalized COVID-19 patients in our study presented more comorbidities compared with patients with COVID-19 undergoing home care, the increased sHLA-G levels were not associated with the occurrence of comorbidities. Although we did not observe the influence of comorbidities on the sHLA-G levels, a study using a larger series of patients may clarify this issue.

Concerning COVID-19 pharmacological treatment, several drugs have been implicated in the risk for COVID-19 susceptibility in patients exhibiting underlying disorders [43]. On the other hand, some drugs used for COVID-19 treatment, such as corticosteroids, have been associated with the modulation of HLA-G expression [44,45,46]. No information is available regarding the role of drugs for COVID-19 treatment on sHLA-G levels. This study demonstrated, for the first time, the influence of the currently available drugs on the treatment of COVID-19 on sHLA-G levels. Although corticosteroids induce the expression of HLA-G, the patients’ treatment with these drugs did not influence their sHLA-G levels. On the other hand, the use of ceftriaxone was associated with increased sHLA-G levels. Since ceftriaxone is not considered to be a transcription factor that may modulate the expression of HLA-G, further studies are needed to unveil this mechanism.

Besides drugs, sexual hormones and cytokines have been reported to modulate the expression of HLA-G [44,47,48]. To understand the reciprocal influence of cytokines and sHLA-G levels, we performed several correlational analyses among these soluble molecules. As observed in this study and others, IL-6, IL-8 and IL-10 have systematically been reported as increased in COVID-19, particularly in the acute phase of the disease [49]. Considering that sHLA-G levels were increased in COVID-19 patients, irrespective of disease severity, the increased sHLA-G levels occurred together with the currently reported inflammatory IL-6, IL-8, and IL-10 cytokines, and positive correlations were observed between sHLA-G and these cytokines; only IL-10 has a well-recognized role on the induction of the *HLA-G* gene expression [50]. In this way, we proposed that sHLA-G levels might be considered a biological marker of the acute phase of COVID-19. In contrast, when we evaluated the risk conferred for the elevated levels of sHLA-G on COVID-19 severity and mortality, no associations were observed.

The increased levels of sHLA-G in the presence of the SARS-CoV-2 infection associated with a worse COVID-19 prognosis and death indicates the deleterious role of HLA-G in viral infections. Finally, this study showed a collection of evidence highlighting the role of the HLA-G molecule in the context of SARS-CoV-2 infection, including (i) sHLA-G levels increased in the presence of SARS-CoV-2 infection *per se*; (ii) sHLA-G levels increased with disease severity but were not associated with patient gender, age, and the presence of comorbidities; (iii) sHLA-G levels decreased in critical COVID-19 phase, suggesting the exhaustion of the immune response; (iv) sHLA-G levels exhibited a positive correlation with the other mediators currently observed in the acute phase of the disease, including IL-6, IL-8 and IL-10; (v) although sHLA-G levels may be associated with an acute biomarker of COVID-19, the increased levels alone were not associated with mortality or with death due to COVID-19. Whether the SARS-CoV-2 *per se* or the innate/adaptive immune response against the virus is responsible for the increased levels of sHLA-G are questions that need to be further addressed [40].

## 4. Materials and Methods

### 4.1. Study Design and Local

This observational, descriptive cross-sectional study, carried out on individuals with COVID-19, was conducted in Ribeirão Preto, State of São Paulo (SP), Brazil. The participants of the healthy control group were from the community of the University of São Paulo (USP). The patients with COVID-19 were from hospital “Santa Casa de Misericórdia de Ribeirão Preto”. The analyses were carried out in the research laboratories of the Faculty of Pharmaceutical Sciences of Ribeirão Preto—USP and of the Medical School of Ribeirão Preto–USP.

### 4.2. Participants

The study enrolled 239 individuals, and the samples were collected from the following stratified groups: (i) healthy controls: 50 participants, from the USP—Campus Ribeirão Preto community, with a negative molecular test of polymerase chain reaction (PCR) for SARS-CoV-2 (post-nasopharyngeal swab) and self-declared healthy and asymptomatic; (ii) COVID-19 patients followed up and treated at home: 60 asymptomatic or oligosymptomatic participants with positive results for their molecular test (PCR) for SARS-CoV-2. These participants came from household searches after consulting reports from the Epidemiological Surveillance of Ribeirão Preto and/or from the Hygia system to identify positive-tested cases, carried out via Basic Health Units (BHUs) in Ribeirão Preto; and (iii) COVID-19 patients admitted for treatment at the hospital comprising 129 participants who exhibited a positive PCR test for SARS-CoV-2. After clinical and laboratory evaluations, patients were classified according to the severity of the disease into asymptomatic/mild, moderate, severe, and critical groups, following the WHO recommendations (Appendix A) [24,25,26,27,28]. Exclusion criteria included the presence of other infectious diseases, age under 18 years old, and pregnancy. The protocol of the study was approved by the local Ethics Research Committee (Process CAAE#30525920.7.0000.5403), and informed consent was obtained from all the participants.

### 4.3. Laboratory Methods

Briefly, 4 mL of peripheral venous blood was collected in Vacutainer tubes (Beckton and Dickinson, Franklin Lakes, NJ, USA) containing EDTA K3 (0.054 mL/tube) to obtain the plasma and carry out the analyses proposed in the study. Data and sample collection occurred from June to December 2020.

### 4.4. Quantification of Soluble HLA-G Plasma Levels

The soluble isoform of HLA-G was detected using an enzyme-linked immunosorbent assay (ELISA). The sHLA-G isoforms were captured by incubating the plasma samples on an immobilized monoclonal antibody (MEM-G/9) monolayer plate (Exbio, Prague, Czech Republic) for 18 h at 4 °C. Non-specific bindings were blocked using 2% albumin in a phosphate-buffered saline (PBS) solution for 30 min. After washing the plates three times with PBS containing 0.05% Tween 20, the plates were incubated for 1 h with the detection antibody conjugated with the HRP-anti-b2-microglobulin peroxidase and with the substrate (ortho-phenyl-*n*-diamino-dihydroxy chloride) (DAKO, Santa Clara, CA). The reaction was stopped by the addition of 1N H_2_SO_4_ and immediately read at 490 nm using an ELISA spectrophotometer (AgileReader™ ELISA Plate Readers, Avans Biotechnology, Taipei City, Taiwan). sHLA-G levels were estimated by a five-point standard curve (12.5–200 ng/mL), using a cell line culture supernatant expressing the M8-HLA-G5 hybridoma. All plasma samples were tested in duplicate.

### 4.5. Cytokine Measurements by Cytometric Bead Array—CBA

The cytokines IL-1β, IL-6, IL-8, IL-10, TNF and IL-12p70 were quantified in the plasma samples of patients and healthy control subjects using the Cytometric Bead Array (CBA) technique, with a Cytometric Bead Kit Array Human Inflammatory Cytokines Kit (BD^®^ Biosciences, San Diego, CA, USA), following the guidelines described by the manufacturer.

### 4.6. Statistical Analyses

Statistical analyses were performed using the GraphPad Prism program (Version 9), adopting a significance level of 5% (alpha = 0.05). The patient data were compared using chi-squared or Fisher’s exact tests for categorical variables. The comparative analyses referring to quantitative variables were performed using the ANOVA non-parametric Mann–Whitney (for two groups) and Kruskal–Wallis (for three or more groups) tests followed by Dunn’s post-test to compare pairs. The correlations between sHLA-G levels and clinical/laboratory/immunological variables were performed using the Spearman test (r), considering the classification of correlation coefficients as <0.4 (weak correlation), >0.4 to <0.5 (moderate), and >0.5 (strong). A binomial logistic regression analysis was performed using the Jamovi software (Version 1.6–2021) to assess the association of sHLA-G levels with severity and mortality in COVID-19. The receptor-operating characteristic (ROC) curves of sHLA-G concentrations were used to predict the severity and mortality of the disease among COVID-19 patients. The area under the curve (AUC) and *p*-values for the significant differences between patients with COVID-19 and the outcomes related to severity and mortality were also evaluated.

## 5. Conclusions

The plasma levels of the immune checkpoint sHLA-G molecule were increased in the SARS-CoV-2 infection and gradually increased according to the clinical severity of COVID-19, reflecting an attempt of the infected host to counterbalance the influence of inflammatory cytokines. On the other hand, in critically ill patients, sHLA-G levels were diminished, possibly reflecting the exhaustion of the immune system, as evidenced by lymphopenia and cytokine dysregulation. Since the increased sHLA-G levels occurred together with the increased levels of inflammatory cytokines, HLA-G may be considered an additional marker of the active phase of the disease. Although increased levels of sHLA-G have been previously reported, this is the first study concomitantly evaluating the impact of several variables that may influence the outcome of the SARS-CoV-2 infection.

## Figures and Tables

**Figure 1 ijms-23-09736-f001:**
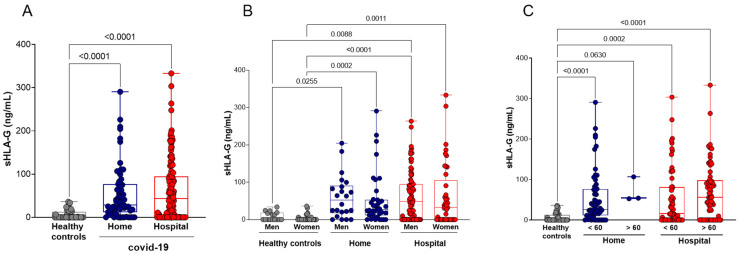
Plasma soluble HLA-G (sHLA-G) levels in COVID-19 patients and healthy controls, stratified by age and gender. (**A**) Healthy controls (*n* = 50) compared with COVID-19 patients undergoing home (*n* = 60) or hospital (*n* = 129) care; (**B**) stratification according to gender: healthy controls (17 men and 33 women), home care COVID-19 patients (22 men and 38 women), and hospital care (87 men and 42 women); (**C**) stratification according to age: healthy controls (50 younger than 60 years), patients undergoing home care (57 younger and 3 older than 60 years), patients undergoing hospital care (59 younger and 70 older than 60 years). Statistical analyses were performed using the Kruskal–Wallis multiple comparison test (non-parametric), followed by Dunn’s post-test. Data are expressed as median and minimum and maximum values. Statistical difference between groups for *p* < 0.05.

**Figure 2 ijms-23-09736-f002:**
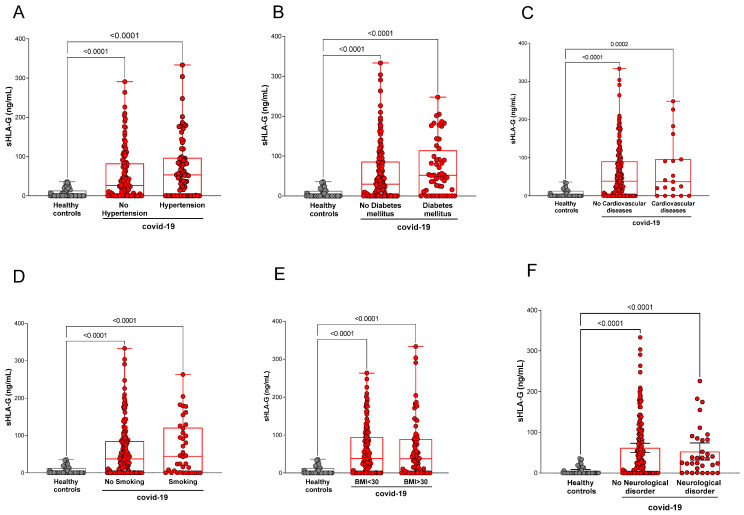
Stratification of plasma sHLA-G levels according to comorbidities. (**A**) Healthy controls (*n* = 50) and COVID-19 patients with (*n* = 76) or without hypertension (*n* = 113); (**B**) healthy controls (*n* = 50) and COVID-19 patients with (*n* = 53) or without (*n* = 136) diabetes mellitus; (**C**) healthy controls (*n* = 50) and COVID-19 patients with (*n* = 19) or without (*n* = 170) cardiovascular diseases; (**D**) non-smoker healthy controls (*n* = 50) and COVID-19 non-smokers (*n* = 151) or smokers (*n* = 38) patients; (**E**) healthy controls exhibiting body mass index (BMI < 30, *n* = 50) and COVID-19 exhibiting BMI < 30 (*n* = 117) or BMI > 30 (*n* = 72); (**F**) healthy controls (*n* = 50) and COVID-19 patients with (*n* = 34) or without neurological disorder (*n* = 155). Statistical analyses were performed using the Kruskal–Wallis multiple comparison test (non-parametric), followed by Dunn’s post-test to compare pairs. Data are expressed as median and minimum and maximum values. Statistical difference between groups for *p* < 0.05.

**Figure 3 ijms-23-09736-f003:**
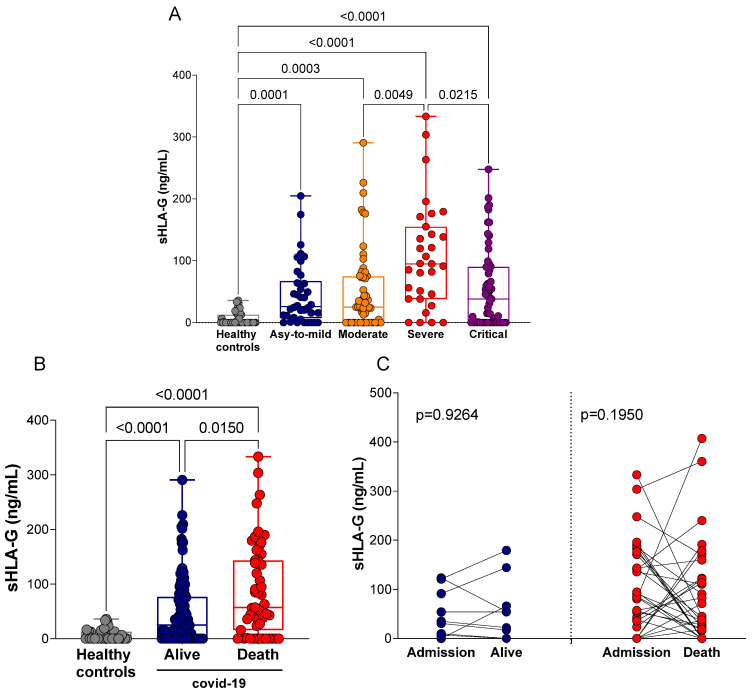
Quantification of plasma sHLA-G levels in hospitalized COVID-19 patients stratified according to disease severity and outcome. (**A**) Healthy controls (*n* = 50) and patients exhibiting asymptomatic/mild (*n* = 39) (Asy-to-mild), moderate (*n* = 56), severe (*n* = 32) and critical (*n* = 62) disease; (**B**) healthy controls (*n* = 50) and patients who were discharged (*n* = 132) and patients who died (*n* = 57) because of COVID-19; (**C**) longitudinal analysis of plasma sHLA-G concentration (ng/mL) at hospital admission (*n* = 10) and at discharge (*n* = 10); and at admission (*n* = 31) compared with death (*n* = 31). Statistical analyses were performed using the Kruskal–Wallis multiple comparison test (non-parametric), followed by Dunn’s post-test to compare pairs. Data are expressed as median and minimum and maximum values. Statistical difference between groups for *p* < 0.05. Longitudinal statistical analysis was performed using the Mann–Whitney test.

**Figure 4 ijms-23-09736-f004:**
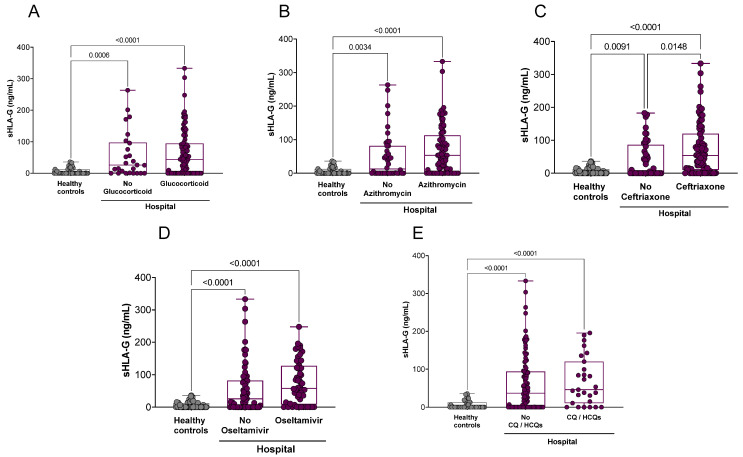
Stratification of plasma sHLA-G levels according to the major pharmacological treatment prescribed to hospitalized COVID-19 patients. (**A**) Treated (*n* = 103) or not (*n* = 26) with glucocorticoids; (**B**) treated (*n* = 86) or not (*n* = 43) with azithromycin; (**C**) treated (*n* = 84) or not (*n* = 45) with ceftriaxone; (**D**) treated (*n* = 52) or not (*n* = 77) with oseltamivir; and (**E**) treated (*n* = 28) or not (*n* = 101) with chloroquine/hydroxychloroquine (CQ/HCQs). Statistical analyses were performed using the Kruskal–Wallis multiple comparison test (non-parametric), followed by Dunn’s post-test to compare pairs. Data are expressed as median and minimum and maximum values. Statistical difference between groups for *p* < 0.05.

**Figure 5 ijms-23-09736-f005:**
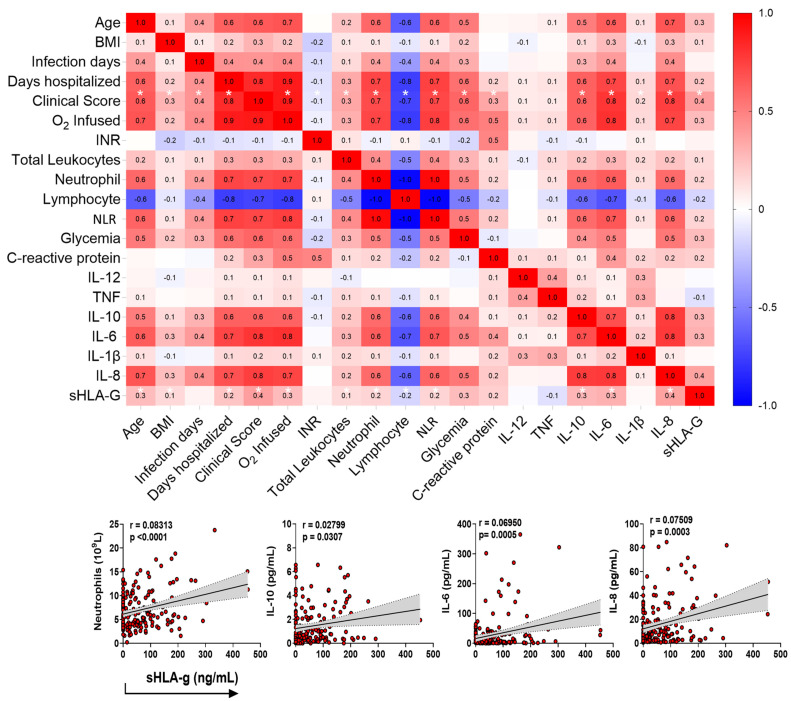
Correlation test matrix between the plasma levels of sHLA-G with the demographics and clinical variables. The upper panel shows the correlation matrix between sHLA-G levels and clinical/immunological variables of COVID-19 patients at home or in hospital care, using the Spearman correlation test (r). The sidebar shows the color scale, which represents positive correlations (red) and negative correlations (blue). Color intensity represents strong correlations. Significant correlations between sHLA-G and clinical score, with *p* < 0.05, are shown with an asterisk (*). The lower panel emphasizes positive correlations primarily associated with the cells and cytokine profiles currently observed in the acute phase of COVID-19.

**Figure 6 ijms-23-09736-f006:**
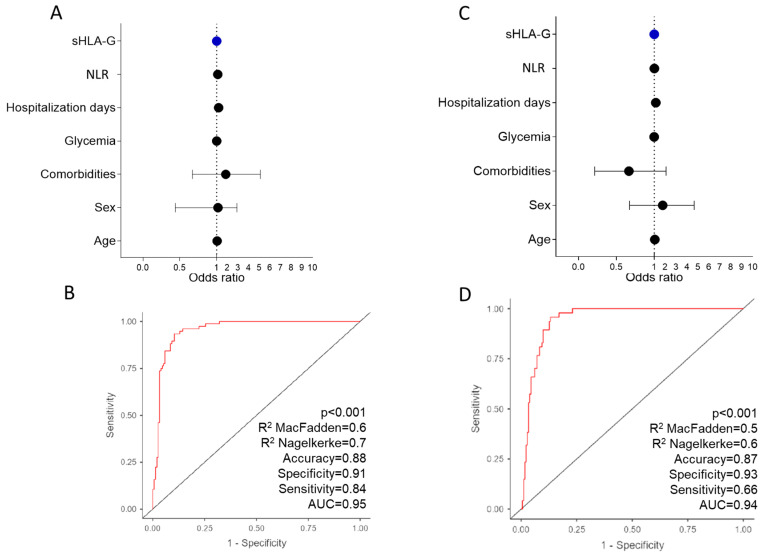
Multivariate analysis of binomial logistic regression regarding the outcome of COVID-19. Evaluated by the disease severity (**A**) and mortality (**C**), encompassing healthy controls *n* = 50, asymptomatic/mild (*n* = 39), moderate (*n* = 56), severe (*n* = 32), and critical (*n* = 62) patients. Receptor operating characteristic (ROC) curves of sHLA-G concentrations were constructed to predict severity (**B**) and mortality (**D**) among COVID-19 patients. The area under the curve (AUC) and *p*-values for significant differences between patients with COVID-19 and the outcome severity and mortality are also shown on the graphic panel. NLR: neutrophil–lymphocyte ratio; OD: odds ratio; CI: confidence interval.

**Table 1 ijms-23-09736-t001:** Demographic, clinical, laboratory, and treatment features of COVID-19 patients and healthy controls.

Variable	Healthy Controls *n* = 50	COVID-19 Patients *n* = 189	^a^*p-*Value	COVID-19 Care		^b^*p-*Value
				Home *n* = 60	Hospital *n* = 129	
**Demographic characteristics**						
Age (mean ± SD)	34.8 ± 9.9	56 ± 19.0	<0.0001	40 ± 13.3	63 ± 16.8	<0.0001
**Gender, *n*. (%)**						
Men	17 (34)	109 (57.7)	0.0029	22 (36.7)	87 (67.4)	<0.0001
Women	33 (66)	80 (42.3)		38 (63.3)	42 (32.6)	
BMI (kg/m^2^, mean ± SD)	25 ± 4.4	27.9 ± 6.7	<0.0001	27.2 ± 5.7	28.3 ± 7.0	0.5646
**Comorbidities, *n*. (%)**						
Hypertension	-	76 (40.2)	-	6 (10.0)	70 (54.3)	<0.0001
Cardiovascular disorder	-	19 (10.1)	-	10 (16.7)	9 (7.0)	0.0392
Diabetes mellitus	-	53 (28.0)	-	8 (13.3)	45 (34.9)	<0.0021
History of smoking	-	38 (20.1)	-	5 (8.3)	33 (25.6)	0.0059
Neurological disorder	-	34 (18.0)	-	17 (28.3)	17 (13.2)	0.0116
**Symptoms, *n*. (%)**						
Dyspnea	-	112 (59.3)	-	23 (38.3)	89 (69.0)	<0.0001
Fever	-	60 (31.7)	-	3 (5.0)	57 (44.2)	<0.0001
Myalgia	-	37 (19.6)	-	-	37 (28.7)	^-^
Diarrhea	-	50 (26.5)	-	26 (43.3)	24 (18.6)	0.0003
Cough	-	130 (68.8)	-	49 (81.7)	81 (62.8)	0.0091
Hyperactive delirium	-	12 (6.4)	-	-	12 (9.3)	^-^
Dysgeusia	-	55 (29.1)	-	42 (70.0)	13 (10.1)	0.0050
Anosmia	-	59 (31.2)	-	42 (70.0)	17 (13.2)	<0.0001
**Laboratory findings** (mean ± SD)						
Erythrocytes × 10^9^/L	4.7 ± 0.5	4.3 ± 0.8	0.0295	4.8 ± 0.5	4.1 ± 0.8	<0.0001
Hemoglobin (g/dL)	14.4 ± 1.4	13 ± 2.4	0.0012	14.6 ± 1.2	12.1 ± 2.5	<0.0001
Leukocytes × 10^9^/L	7.6 ± 2.0	9.5 ± 4.9	0.0498	7.7 ± 2.3	10.4 ± 5.6	0.0038
Neutrophils × 10^9^/L	4.4 ± 1.6	7.3 ± 4.6	<0.0001	4.5 ± 2.0	8.6 ± 4.9	<0.0001
Lymphocytes × 10^9^/L	2.4 ± 0.7	1.5 ± 1.0	<0.0001	1.5 ± 0.9	1.1 ± 0.6	0.0001
Neutrophil/lymphocyte ratio	1.8 ± 0.6	7.2 ± 6.8	<0.0001	6 ± 5.9	8.6 ± 7.6	0.0173
Monocytes × 10^9^/L	0.5 ± 0.2	0.5 ± 0.3	>0.1	0.5 ± 0.2	0.5 ± 0.4	>0.1
Platelets × 10^9^/L	225.5 ± 47.1	252.5 ± 98.7	0.9818	233.5 ±68.6	261.4 ± 109	>0.1
**Hospital care, *n*. (%)**						
Ward	-	94 (49.7)	-	-	94 (72.9)	^-^
Intensive care unit	-	35 (18.5)	-	-	35 (27.1)	^-^
**Hospitalization data** (mean ± SD)						
Days in hospital	-	10.6 ± 7.3	-	-	10.6 ± 7.3	-
Days from symptom onset to recruitment	-	4.9 ± 3.6	-	6.7 ± 2.9	4.0 ± 3.6	0.0004
**Respiratory support received, (%)**						
High flow nasal cannula	-	60 (31.7)	-	-	60 (46.5)	^-^
Oxygen masks/Non-invasive	-	33 (17.5)	-	-	33 (25.6)	^-^
Invasive ventilation	-	27 (14.3)	-	-	27 (20.9)	^-^
Oxygen Saturation (mean ± SD)	98.3 ± 1.5	91.8 ± 8.0	<0.0001	97.4 ± 2.0	89.4 ± 8.4	<0.0001
**Medications, *n*. (%)**						
Glucocorticoid	-	103 (54.5)	-	-	103 (79.4)	^-^
Azithromycin	-	86 (45.5)	-	-	86 (66.7)	^-^
Ceftriaxone	-	84 (44.4)	-	-	84 (65.1)	^-^
Oseltamivir	-	52 (27.5)	-	-	52 (40.3)	^-^
CQ/HCQs	-	28 (14.8)	-	-	28 (21.7)	^-^
Anticoagulant	-	9 (4.8)	-	1 (1.7)	8 (6.2)	0.1730
Ivermectin	-	1 (0.5)	-	-	1 (0.8)	^-^

Abbreviations: Data are expressed as mean, SD: standard deviation; BMI: body mass index; CQ/HCQs: chloroquine/hydroxychloroquine sulfate; ^a^
*p*-value: comparisons between the healthy controls vs. COVID-19 patients; ^b^
*p*-value: comparisons between the COVID-19 patients in home care vs. hospital. Significant differences were assessed using the Mann–Whitney U test (to compare two groups) or using a chi-squared test (categorical variables). *p* < 0.05 was considered statistically significant.

## Data Availability

Not applicable.

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
