# Peer review of "The Severity of COVID-19 Affects the Plasma Soluble Levels of the Immune Checkpoint HLA-G Molecule"

_ijms, 2022, doi:10.3390/ijms23179736_

Round 1

Reviewer 1 Report

Dear Authors,

The manuscript entitled "Severity of COVID-19, but not pharmacological interventions or comorbidities of patients, affects levels of plasma soluble HLA-G: an immunoinhibitory molecule" presented data with some value regarding the levels of serum HLA-G and its relation with the COVID-19 severity in patients. However, major revisions are required before the submitted manuscript to be further processed for the current journal. Below you can find my comments.

1) The title is too big and requires revision, to attract more the attention of the readers.

2) The whole manuscript suffers from serious language mistakes. To improve the language quality, please use a native English speaker to perform the language polishing.

3) 2. Results section - Table 1.

A. The table 1 is a little misconfused. The demographic characteristics include only age as an absolute number right? Ι do not understand what the column "All patients" is representing exactly. If you make the calculations of healthy controls, Residential and Hospitalized this number is 262 patients and not 189. Please adjust more specific you sample data. Also, this column should be placed at the right of the table 1 after the "Hospitalized" and before the p-value. 

B. In addition the other question that i have is the group of healthy controls, how healthy are? Since there is a number of subjects in health controls that suffers from hypertension, cardiovascular disease, diabetes Mellitus, these disorders may affect the levels of sHLA-G and also the levels of the detected inflammatory cytokines. For this reason, these subjects must be excluded. To be more specific also a totally healthy control group must be added without any underlying disorders. In this group, the sHLA-G and inflammatory cytokines must be detected, and this could lead to the identification of different levels of sHLA-G and inflammatory cytokines, either between "healthy individuals".

C. When patients are used initially an informed consent must be written signed by all. Also, the whole study needs approval by the bioethics committee, and should be conformed according to the directions of Helsinki declaration. However, neither of the above i was able to found in the submitted article.

4) The statistics in all figures require reconsidering. The HLA-G is playing crucial role in immunoregulation. However, its increase in COVID-19 is mostly related not with the acute phase of the disease, rather than when the disease, is passing out. For this purpose i am not sure the HLA-G as its can be used as a potential biomarker as the authors intended to. On the other hand to be used alongside with the proinflammatory cytokine profile of the patient is something that can be discussed. But this must be supported also by more results in this study.

5) To support more the results of your study, sequencing of HLAs from each individual, should be performed. Additional, flow cytometric analysis of peripheral blood subpopulations is required, in order to detect the activate or non activate form of macrophages, T and B cells from all individuals.\

6) In the discussion section more studies need to be added, in order to compare better the results.

Author Response

Reviewer #1

The manuscript entitled "Severity of COVID-19, but not pharmacological interventions or comorbidities of patients, affects levels of plasma soluble HLA-G: an immunoinhibitory molecule" presented data with some value regarding the levels of serum HLA-G and its relation with the COVID-19 severity in patients. However, major revisions are required before the submitted manuscript is further processed for the current journal. Below you can find my comments.

Answer: Thank you for the careful revision of our manuscript. See below the comments and answers to your queries:

  1. The title is too big and requires revision, to attract more the attention of the readers.

Answer: As per your suggestion, the title of the article was shortened as follows: “The severity of covid-19 affects the plasma soluble levels of the immune checkpoint HLA-G molecule

  1. The whole manuscript suffers from serious language mistakes. To improve the language quality, please use a native English speaker to perform the language polishing.

Answer: The manuscript was completely revised for English style, grammar, and spelling mistakes. The introduction and the Discussion sections were entirely revised and many phrases were rephrased to clarify the major ideas.

  1. Results section - Table 1.
  2. Table 1 is a little confusing. The demographic characteristics include only age as an absolute number right? Ι do not understand what the column "All patients" is representing exactly. If you make the calculations of healthy controls, Residential and Hospitalized this number is 262 patients and not 189. Please adjust more specific sample data. Also, this column should be placed at the right of table 1 after the "Hospitalized" and before the p-value.

Answer: We studied a total of 239 individuals; i.e., 189 patients with covid-19 (129 admitted to a hospital and 60 followed-up at home) and 50 healthy controls (we excluded control individuals exhibiting hypertension, diabetes, cardiac disorders, and the smoker ones), and all analyses were reevaluated. Then, Table 1 and the mentions of these numbers in the Material and Methods sections were clarified and corrected in the revised version of the manuscript.

  1. In addition the other question that I have is the group of healthy controls, how healthy is they? Since several subjects in health controls suffers from hypertension, cardiovascular disease, and diabetes Mellitus, these disorders may affect the levels of sHLA-G and also the levels of the detected inflammatory cytokines. For this reason, these subjects must be excluded. To be more specific also a healthy control group must be added without any underlying disorders. In this group, the sHLA-G and inflammatory cytokines must be detected, and this could lead to the identification of different levels of sHLA-G and inflammatory cytokines, either between "healthy individuals".

Answer: As previously stated, healthy individuals exhibiting comorbidities were excluded, yielding a total of 50 individuals. All analyses were reevaluated accordingly. 

  1. When patients are used initially informed consent must be written and signed by all. Also, the whole study needs approval by the bioethics committee and should be conformed according to the directions of Helsinki declaration. However, neither of the above I was able to found in the submitted article.

Answer: In the original version of the manuscript, the information regarding the approval by Ethics Research Committee was mentioned in the footnote at the end of the manuscript, as recommended by the Journal rules. Additionally, this information was also included in the Material and Methods section of the revised version of the manuscript, as follows The protocol of the study was approved by the local Ethics Research Committee (Process CAAE nº 30525920.7.0000.5403), and informed consent was obtained from all participants”.

  1. The statistics in all figures require reconsidering. The HLA-G is playing a crucial role in immunoregulation. However, its increase in COVID-19 is mostly related not to the acute phase of the disease, but rather to when the disease, is passing out. For this purpose, I am not sure about the HLA-G as it can be used as a potential biomarker as the authors intended to. On the other hand, being used alongside the proinflammatory cytokine profile of the patient is something that can be discussed. But this must be supported also by more results in this study.

Answer: The idea of including HLA-G as a potential biomarker for disease activity has arisen from the following results, obtained in this study: i) HLA-G was increased in covid-19 patients, irrespective of the severity of the disease for patients followed-up at home or hospital care, ii) HLA-G was increased together with the currently reported inflammatory IL-6, IL-8, and IL-10 cytokines, and iii) positive correlations were observed between IL-6 and sHLA-G (r=0.4), IL-8 and sHLA-G (r=0.3), and IL-10 with sHLA-G (r=0.3). Therefore, we agree with your idea that HLA-G may be considered as a marker of disease activity along with the classical inflammatory cytokines, characteristics of covid-19 activity, rather than a biomarker for covid-19. These comments were included in the discussion section of the revised version of the manuscript.

  1. To support the results of your study, sequencing of HLAs from each individual should be performed. Additional, flow cytometric analysis of peripheral blood subpopulations is required, in order to detect the activate or non-activate form of macrophages, T and B cells from all individuals.

Answer: Indeed, the sequencing analyses of the HLA-G gene, encompassing the regulatory and coding regions would be of great interest, since several promoter regions and 3' UTR haplotypes have been associated with the differential production of sHLA-G. Although we have collected DNAs of all these patients, at this moment: i) it would be impossible to perform all these sequencings within the short time to review the manuscript, and ii) this issue would be an interesting topic for another investigation since only the 14 base pair polymorphism was evaluated in terms of covid-19 susceptibility. Noteworthy, these HLA-G genotypes were not associated with the sHLA-G levels in covid-19 patients (Ad’hiah et al 2022). On the other hand, the evaluation of the macrophage activation and T and B lymphocyte subpopulations is also a very interesting issue; however, we would need a new cohort of covid-19 patients, since no cells have been collected and stored from all these patients. We thank you for these suggestions, and we hope that the Referee could understand our reasons.  

  1. In the discussion section more studies need to be added, to compare better the results.

Answer: The Introduction and Discussion sections were completely remodeled to encompass most of the articles that have already been published regarding HLA-G and covid-19, contributing to improving the discussion of our results.

Reviewer 2 Report

The manuscript by Correa Cordeiro et al., entitled "Severity of COVID-19, but not pharmacological interventions or comorbidities of patients, affects levels of plasma soluble HLA-G: an immunoinhibitory molecule" is written in plain English, causing sentences with unclear content, especially in the introduction part. Language needs work in mostly grammar, less wording (e.g. t<ble 1, text, fig. 3A).

In the introduction, most information about HLA-G remain unclear and seem contradictory. Some published articles, especially about HLA-G in COVID patients are not mentioned. The major topic, HLA-G, is only named by its abbreviation, at least it should be named completely in introduction.

In results, the number of COVID-patients included differ from the number mentioned in the abstract; it might be a language problem in the abstract. Please correct.

The control-group does not match concerning age in respect to patient groups.

In the main table in results, p-values are given for all characteristics, even for age or gender referring to regression analysis but not further discussed. Different statistical tests are mentioned to be used in each data group. It remains open which exact test was used. Timepoint of measuring HLA-G levels in patients are not mentioned (beginning/middle/end of disease, median?) even for patients reported to have died (before/after death?). Regressions were calculated as well for HLA-G levels. Figure 4 shows regression analysis between control group and patient’s groups with or with pharmacological treatments, no analysis was done between the patient’s groups.

In discussion, results are demonstrated some without putting them in relation to published data and there are chapters (>50%) describing published data without relation to obtained data. One chapter mentions age as an important factor, but in results, no correlation had been done in the age group of above 60y, due to low number of patients in the control group (as it is not discussed anywhere as a drawback of the study). Discussion needs to be rewritten concerning obtained results (e.g. of fig. 5) and relation to published data.

In the M&M section, participants are described to be minimum 18y. In table 1, age starts at 16y. Please adopt and recalculate numbers and regressions of all values.

After revision I would be willing to review the manuscript again.

Author Response

The authors thank Reviewer #2 for the positive evaluation of our manuscript.

  1. The manuscript by Correa Cordeiro et al., entitled "Severity of COVID-19, but not pharmacological interventions or comorbidities of patients, affects levels of plasma soluble HLA-G: an immunoinhibitory molecule" is written in plain English, causing sentences with unclear content, especially in the introduction part. Language needs work in mostly grammar, less wording (e.g. t<ble 1, text, fig. 3A).

Answer: The manuscript was revised for English style, grammar, and spelling mistakes. The Introduction and Discussion sections were completely revised, clarified, and shortened, eliminating repetitive information. Tables were also revised, and Table and Figure legends were simplified.    

  1. In the introduction, most information about HLA-G remain unclear and seem contradictory. Some published articles, especially about HLA-G in COVID patients are not mentioned. The major topic, HLA-G, is only named by its abbreviation, at least it should be named completely in introduction.

Answer: As mentioned in the above query, phrases in the Introduction section were rephrased. The HLA abbreviation was also included in the Introduction section. Additionally, we included several lines of evidence reported in the literature regarding the role of HLA-G in covid-19, as follows: “Several lines of evidence support the role of HLA-G in covid-19 pathogenesis, including: i) increased levels of sHLA-G molecules in patients exhibiting various clini-cal manifestations of the disease [14–18], ii) increased number of immune system cells expressing HLA-G [16,19], iii) association of the HLA-G 3’untranslated region poly-morphism with susceptibility to the disease [18], iv) increased tissue expression of HLA-G in case reports [20,21], and v) induction of the expression of hub genes, includ-ing HLA-G, in lung tissue infected by SARS-CoV-2 [22]”.

  1. As result, the number of COVID patients included differs from the number mentioned in the abstract; it might be a language problem in the abstract.

Answer: We clarified the information regarding the number of patients and controls in the Abstract, Material and Methods section, and Tables. As per suggestion of Reviewer#1, we eliminated all controls individuals exhibiting comorbidities, and all results were reanalyzed accordingly. As a consequence, the number of healthy controls decreased to 50 individuals.  

  1. The control group does not match the age of the patient groups.

Answer: Indeed, the covid-19 patients were older than the healthy control group and the major reason accounts for the difficulty in recruiting older healthy individuals, particularly, during the pandemics, when all samples were collected. Additionally, recruiting older healthy subjects during lockdown periods would put in risk these individuals. Therefore, we selected healthy University Hospital staff that needed to work during the pandemics. To circumvent the possible influence of the age difference, we performed several multivariate analyses to rule out the influence of age on the studied variables. Fortunately, age (<60 or > 60 years) did not influence the variables evaluated in this study.

  1. In the main table of results, p-values are given for all characteristics, even for age or gender referring to regression analysis but not further discussed. Different statistical tests are mentioned to be used in each data group. It remains open which exact test was used.

Answer: First, the procedures for statistical analyses written in the original form of the manuscript were rewritten and clarified at the Material and Methods section. We further included the major statistical analyses in all Table and Figure legends. The regression analyses referring to the influence of age and gender was rephrased and mentioned at the Results and Discussion sections. 

  1. The time point of measuring HLA-G levels in patients is not mentioned (beginning/middle/end of disease, median?) even for patients reported having died (before/after death?). Regressions were calculated as well for HLA-G levels. Figure 4 shows regression analysis between the control group and patient groups with or with pharmacological treatments, no analysis was done between the patient groups.

Answer: The information regarding the time points of blood collection for sHLA-G quantification were introduced in Table 1. The results referring to the regression analyses concerning all variables studied, including pharmacological treatment were discussed in the revised version of the manuscript. The comparisons between the pharmacological treatment among patients groups was not performed since patients treated at home used only antipyretic and analgesic drugs, whereas hospitalized used several types of drugs, and the sHLA-G levels were compared only between hospitalized patients using or not specific drugs. This information was clarified in the revised version of the manuscript.

  1. In discussion, results are demonstrated some without putting them concerning published data and there are chapters (>50%) describing published data without relation to obtained data. One chapter mentions age as an important factor, but in the results, no correlation had been done in the age group of above 60y, due to low number of patients in the control group (as it is not discussed anywhere as a drawback of the study). Discussion needs to be rewritten concerning obtained results (e.g. of fig. 5) and relation to published data.

Answer: The Introduction and Discussion sections were completely modified, including all reported evidence of the participation of HLA-G in covid-19, including several references that were not previously mentioned. 

  1. In the M&M section, participants are described to be a minimum 18y. In table 1, the age starts at 16y. Please adopt and recalculate numbers and regressions of all values.

Answer: We apologize for this typing error, only individuals older than 18 years were studied.

  1. After revision I would be willing to review the manuscript again.

Answer: We hope that the revised version of the manuscript can reach your major concerns.

Round 2

Reviewer 1 Report

Dear Authors,

You have well addressed the majority of my concerns. Well done!!!

Reviewer 2 Report

The revised manuscript by Correa Cordeiro et al., entitled (new title) "The severity of covid-19 affects the plasma soluble levels of the immune checkpoint HLA-G molecule" improved sufficiently by made corrections and additional text passages.

Authors did respond well to all raised questions. Spelling and grammar were revised accordingly.

I therefore can recommend the manuscript now for publication.

This manuscript is a resubmission of an earlier submission. The following is a list of the peer review reports and author responses from that submission.